# Association of Serum Albumin Levels and Long-Term Prognosis in Patients with Biopsy-Confirmed Nonalcoholic Fatty Liver Disease

**DOI:** 10.3390/nu15092014

**Published:** 2023-04-22

**Authors:** Hirokazu Takahashi, Miwa Kawanaka, Hideki Fujii, Michihiro Iwaki, Hideki Hayashi, Hidenori Toyoda, Satoshi Oeda, Hideyuki Hyogo, Asahiro Morishita, Kensuke Munekage, Kazuhito Kawata, Tsubasa Tsutsumi, Koji Sawada, Tatsuji Maeshiro, Hiroshi Tobita, Yuichi Yoshida, Masafumi Naito, Asuka Araki, Shingo Arakaki, Takumi Kawaguchi, Hidenao Noritake, Masafumi Ono, Tsutomu Masaki, Satoshi Yasuda, Eiichi Tomita, Masato Yoneda, Akihiro Tokushige, Yoshihiro Kamada, Shinichiro Ueda, Shinichi Aishima, Yoshio Sumida, Atsushi Nakajima, Takeshi Okanoue

**Affiliations:** 1Liver Center, Saga Medical School, Saga University, Saga 849-8501, Japan; takahas2@cc.saga-u.ac.jp (H.T.); ooedasa@cc.saga-u.ac.jp (S.O.); 2Department of General Internal Medicine2, Kawasaki Medical Center, Okayama 700-8505, Japan; m.kawanaka@med.kawasaki-m.ac.jp; 3Department of Hepatology, Graduate School of Medicine, Osaka Metropolitan University, Osaka 558-8585, Japan; 4Division of Gastroenterology and Hepatology, Yokohama City University Graduate School of Medicine, Yokohama 236-0004, Japan; michihirokeidai@yahoo.co.jp (M.I.);; 5Department of Gastroenterology and Hepatology, Gifu Municipal Hospital, Gifu 500-8323, Japan; hide-hayashi@umin.ac.jp (H.H.); etomita_jp@yahoo.co.jp (E.T.); 6Department of Gastroenterology, Ogaki Municipal Hospital, Ogaki 503-8502, Japan; hmtoyoda@spice.ocn.ne.jp (H.T.); satoshi.yasuda.1982@gmail.com (S.Y.); 7Department of Laboratory Medicine, Saga University Hospital, Saga 849-8501, Japan; 8Hyogo Life Care Clinic Hiroshima, Hiroshima 732-0823, Japan; hidehyogo@ae.auone-net.jp; 9Department of Gastroenterology and Neurology, Faculty of Medicine, Kagawa University, Kagawa 761-0793, Japan; asahiro@med.kagawa-u.ac.jp (A.M.); tmasaki@med.kagawa-u.ac.jp (T.M.); 10Department of Gastroenterology and Hepatology, Kochi Medical School, Kochi 783-8505, Japan; jm-k.munekage@kochi-u.ac.jp; 11Hepatology Division, Department of Internal Medicine II, Hamamatsu University School of Medicine, Shizuoka 431-3192, Japan; kawata@hama-med.ac.jp (K.K.); noritake@hama-med.ac.jp (H.N.); 12Division of Gastroenterology, Department of Medicine, Kurume University School of Medicine, Kurume 830-0011, Japan; tsutsumi_tsubasa@med.kurume-u.ac.jp (T.T.); takumi@med.kurume-u.ac.jp (T.K.); 13Liver Disease Care Unit, Division of Metabolism and Biosystemic Science, Gastroenterology, and Hematology/Oncology, Department of Medicine, Asahikawa Medical University, Asahikawa 078-8802, Japan; k-sawada@asahikawa-med.ac.jp; 14First Department of Internal Medicine, University of the Ryukyus Hospital, 207 Uehara, Nishihara, Nakagami, Okinawa 903-0215, Japan; f994908@med.u-ryukyu.ac.jp (T.M.); h052010@med.u-ryukyu.ac.jp (S.A.); 15Department of Hepatology, Shimane University Hospital, 89-1 Enya-cho, Izumo 693-8501, Japan; ht1020@med.shimane-u.ac.jp (H.T.); asuka@med.shimane-u.ac.jp (A.A.); 16Department of Gastroenterology and Hepatology, Suita Municipal Hospital, Osaka 564-8567, Japan; yu1yoshida@gmail.com (Y.Y.); naito0757@mhp.suita.osaka.jp (M.N.); 17Division of Innovative Medicine for Hepatobiliary & Pancreatology, Faculty of Medicine, Kagawa University, Kita 761-0793, Japan; ono.masafumi@kagawa-u.ac.jp; 18Department of Clinical Pharmacology and Therapeutics, Graduate School of Medicine, University of the Ryukyus, Okinawa 903-0215, Japan; 19Department of Advanced Metabolic Hepatology, Osaka University Graduate School of Medicine, 1-7, Yamadaoka, Suita 565-0871, Japan; 20Department of Pathology and Microbiology, Faculty of Medicine, Saga University, Saga 849-8501, Japan; saish@cc.saga-u.ac.jp; 21Division of Hepatology and Pancreatology, Department of Internal Medicine, Aichi Medical University, Nagakute 480-1195, Japan; 22Hepatology Center, Saiseikai Suita Hospital, Suita 564-0013, Japan

**Keywords:** nonalcoholic steatohepatitis, albumin, prognosis, Asian

## Abstract

The relationship between baseline serum albumin level and long-term prognosis of patients with nonalcoholic fatty liver disease (NAFLD) remains unknown. This is a sub-analysis of the CLIONE (Clinical Outcome Nonalcoholic Fatty Liver Disease) study. The main outcomes were: death or orthotopic liver transplantation (OLT), liver-related death, and liver-related events (hepatocellular carcinoma [HCC], decompensated cirrhosis, and gastroesophageal varices/bleeding). 1383 Japanese patients with biopsy-confirmed NAFLD were analyzed. They were divided into 3 groups based on serum albumin: high (>4.0 g/dL), intermediate (3.5–4.0 g/dL), and low (<3.5 g/dL). Unadjusted hazard ratio [HR] of the intermediate albumin group, compared with the high albumin group, were 3.6 for death or OLT, 11.2 for liver-related death, 4.6 for HCC, 8.2 for decompensated cirrhosis, and 6.2 for gastroesophageal varices (all risks were statistically significant). After adjusting confounding factors, albumin remained significantly associated with death or OLT (intermediate vs. high albumin group: HR 3.06, 95% confidence interval [CI] 1.59–5.91, *p* < 0.001; low vs. high albumin group: HR 22.9, 95% CI 8.21–63.9, *p* < 0.001). Among biopsy-confirmed NAFLD patients, those with intermediate or low serum albumin had a significantly higher risk of death or OLT than those with high serum albumin.

## 1. Introduction

Nonalcoholic fatty liver disease (NAFLD) is the most common etiology of chronic liver disease, affecting approximately 25% of the adult population worldwide [1]. NAFLD is a complication of obesity, and some patients are thought to progress to nonalcoholic steatohepatitis (NASH), which is characterized histologically by the presence of steatosis, inflammation, and hepatocellular ballooning with or without fibrosis, ultimately leading to cirrhosis, hepatocellular carcinoma (HCC), and death [2,3].

Albumin is the most abundant plasma protein and plays a key role in the regulation of plasma colloid osmotic pressure [4,5]. It also has various other physiologic functions, including solubilization, binding, and transport of endogenous and exogenous molecules; antioxidative, anti-inflammatory, and hemostatic effects; endothelial stabilization; and adjustment of capillary permeability [4,5]. Importantly, albumin is a major prognostic factor in patients with liver cirrhosis, being reported as a significant predictor of death in over 100 studies [6]. It is also a component of the most important and widely used prognostic score in cirrhosis, the Child–Turcotte–Pugh (CTP) score [7].

Hypoalbuminemia is frequently observed in patients with advanced cirrhosis [8] and is generally defined as an intravascular albumin level < 3.5 g/dL [9,10]. The nutritional therapy algorithm in the 2020 guidelines from the Japanese Society of Gastroenterology/Japanese Society of Hepatology recommends conducting a nutritional assessment for hypoalbuminemia, Child–Pugh class B or C, and sarcopenia, each of which exerts an adverse impact on clinical outcomes in patients with cirrhosis [10]. Additionally, Angulo et al. reported that the NAFLD fibrosis score (NFS) could help identify patients with NAFLD at increased risk for liver-related complications or death [11]. This noninvasive test includes serum albumin levels, which indirectly reflect hepatic synthetic reserve [11,12].

Based on these data, we hypothesized that serum albumin levels at the time of liver biopsy could predict the long-term prognosis of patients with NAFLD. Therefore, we used data from a multicenter registry to conduct a cohort study examining the prognostic value of serum albumin in patients with biopsy-confirmed NAFLD.

## 2. Materials and Methods

### 2.1. Study Design and Population

The current study is a sub-analysis of the longitudinal multicenter cohort study called CLIONE (Clinical Outcome Nonalcoholic Fatty Liver Disease) in Asia [13]. The method used to establish the cohort and the primary findings of the CLIONE study were described elsewhere [13]. This investigation was conducted in accordance with the 1964 Declaration of Helsinki and was authorized by the institutional review board of Saga University Hospital (approval no. 2020-04-R-02, 30 June 2020). Informed consent was not deemed necessary since the study employed pre-existing data. This study followed the Strengthening the Reporting of Observational Studies in Epidemiology (STROBE) reporting guidelines.

### 2.2. Data Sources

In this study, we used a database from the Japan Study Group of NAFLD (JSG-NAFLD) to obtain information regarding patients with biopsy-proven NAFLD [13]. All data were managed using REDCap electronic data capture tools, hosted at the Osaka Metropolitan University.

### 2.3. Study Cohort

We identified 1760 patients who underwent liver biopsy for suspected fatty liver between 1 December 1994 and 31 December 2020 [13]. The exclusion criteria were a history of other hepatic diseases, such as viral hepatitis and a history of alcohol abuse (defined as >30 g/day in men; >20 g/day in women). The inclusion criteria were (1) positive serology for HCV but negative for HCV-RNA and no history of treatment, and (2) burned-out NASH (no increased fat and fibrosis stage 4). We used a total of 1398 patients in CLIONE study [13]. In this study, we also excluded 15 patients with missing serum albumin data, and finally, enrolled 1383 Japanese patients with biopsy-confirmed NAFLD. We followed this cohort until 31 March 2021 to identify clinically important out comes: death or OLT, liver-related death, and liver-related events (HCC, decompensated cirrhosis, or gastroesophageal varices/bleeding).

### 2.4. Clinical Assessment

Data were extracted regarding BMI, blood pressure, daily alcohol intake, smoking habits, past medical history, and current drug history. We also recorded plasma glucose, lipids, and liver biochemistry values, which were measured in venous blood samples obtained after fasting for ≥8 h. DM, hypertension, and dyslipidemia were diagnosed according to standard criteria [14,15,16]. FIB-4 and NFS were calculated using the available parameters [12,17]. For this study, we divided the patients into 3 groups according to albumin level: high (>4.0 g/dL), intermediate (3.5–4.0 g/dL), and low (<3.5 g/dL).

### 2.5. Liver Histology

Ultrasound-assisted percutaneous liver biopsies were performed. Liver sections were embedded in paraffin after being fixed in formalin and stained with either hematoxylin and eosin or azan. These sections were then sent to an experienced pathologist (S.A.) at Saga University for central evaluation, who was unaware of the patients’ clinical and laboratory records. NAFLD was defined as the presence of ≥5% hepatic steatosis (according to Kleiner et al. [18]). Grading and staging were performed according to Brunt et al. [19] and Kleiner et al. [18]. Advanced fibrosis was defined as stage 3 or 4 fibrosis. NASH was diagnosed according to the fatty liver inhibition of progression (FLIP) algorithm [20].

### 2.6. Follow-Up Evaluation

Details regarding follow-up evaluations were described previously [13] and are summarized briefly here. The follow-up period began on the day of liver biopsy and continued until the last visit, death, or OLT. Patients were followed at 3- to 12-month intervals after NAFLD diagnosis, and anthropometric measurements and metabolic assessments were repeated during each visit. For liver-related events (HCC, gastroesophageal varices, and decompensated cirrhosis), only the first event after liver biopsy was recorded; recur-rent events were excluded. Follow-up duration was the period between the date of the biopsy and the date of the most recent follow-up. Patient information was collected on an ongoing basis due to the fact that all events took place at the same facility where the patients were being treated. For hospitalizations, we documented the diagnosis given at the time of admission. The duration of follow-up was calculated as the period between the biopsy date and the most recent follow-up. In cases where the date or details of an event in the database were unclear, the lead investigator reviewed the facility records and made the necessary amendments.

### 2.7. Statistical Analysis

Continuous and ordinal variables are expressed as mean (standard deviation) or median (range) and were compared using the unpaired t-test. Categorical variables were compared using the χ^2^ test. Clinically important outcomes are presented as Kaplan–Meier curves, and albumin levels were compared using the log-rank test. Univariate (unadjusted) and multivariate (adjusted) hazard ratio (HR) estimates (relative risk) of clinically significant outcomes were calculated using Cox proportional hazard regression analysis to control for the effects of potential risk factors, while considering follow-up duration. In this study, we created 3 models. Model 1 included serum albumin level, age (binarized as ≥65 years and <65 years), and sex (male or female). Model 2 included variables from model 1 plus NASH (based on the FLIP algorithm) and fibrosis stage. Model 3 included variables from model 1 plus DM and BMI (binarized as ≥25 kg/m^2^ and <25 kg/m^2^). Two-sided *p* values < 0.05 were considered significant. All statistical tests were conducted using JMP^®^ 16.0.0 software (SAS Institute Inc., Cary, NC, USA).

## 3. Results

### 3.1. Baseline Characteristics

The baseline characteristics of the cohort (N = 1383) are presented in Table 1. Mean age was 54.6 years, mean BMI was 27.9 kg/m^2^, and females accounted for 57.1% (*n* = 790) of the cohort. The majority (66.9%) of participants had NASH, and 223 (16%) had advanced fibrosis (stages 3–4). When patients were divided into three categories by albumin levels, we observed a stepwise increase in mean age, aspartate aminotransferase, FIB-4, NFS, and prevalence of advanced fibrosis as albumin levels decreased. Conversely, platelet counts decreased significantly with decreasing albumin levels.

The relationship between fibrosis stage and albumin levels is shown in Figure 1. The mean values of albumin in stages 0, 1, 2, 3, and 4 were 4.5, 4.4, 4.3, 4.2, and 3.8 g/dL, respectively. When albumin levels were analyzed in relation to the histologic stage of fibrosis, the distribution of albumin levels differed by histologic stage (Kruskal–Wallis test, *p* < 0.001).

Data for clinically important outcomes are presented in Table 2. During a median follow-up of 4.5 years (range, 0.3–21.6 years), there were 46 deaths or orthotopic liver transplantation (OLT) and 20 deaths were liver-related. The occurrence of liver-related events was as follows: HCC, 36 patients; decompensated cirrhosis, 23 patients; and gastroesophageal varices, 17 patients.

### 3.2. Clinically Important Outcomes According to Albumin Level

Overall mortality and liver-related events stratified according to albumin are shown in Figure 2. Death or OLT, liver-related death, HCC, decompensated cirrhosis, and gastroesophageal varices differed significantly according to the serum albumin category (high, intermediate, or low; log-rank *p* < 0.001 for all outcomes).

Next, we calculated the unadjusted hazard risks of clinically important outcomes (Figure 3). Among patients with biopsy-proven NAFLD, the hazard risks were significantly increased in the intermediate albumin group, compared with the high albumin group, for death or OLT (3.6), liver-related death (11.2), HCC (4.6), decompensated cirrhosis (8.2), and gastroesophageal varices (6.2). The hazard risks were also higher in the low albumin group, compared with the high albumin group, for death or OLT (20.9), liver-related death (83.6), HCC (4.2), decompensated cirrhosis (45.4), and gastroesophageal varices (38.4). These low albumin group risks were statistically significant for all outcomes except HCC.

Univariate HRs for death or OLT and liver-related death are shown in Appendix A. Lower albumin levels were significantly associated with death or OLT (intermediate vs. high albumin group: HR 3.58, 95% confidence interval [CI] 1.91–6.69, *p* < 0.001; low vs. high albumin group: HR 20.9, 95% CI 7.72–56.8, *p* < 0.001) and liver-related death (intermediate vs. high albumin group: HR 11.2, 95% CI 3.88–32.4, *p* < 0.001; low vs. high albumin group: HR 83.6, 95% CI 21.1–330, *p* < 0.001). Older age (≥65 y) was significantly associated with death or OLT (HR 2.58, 95% CI 1.43–4.68, *p* = 0.002). More advanced fibrosis stage was also significantly associated with liver-related death (stage 3 vs. stage 0; HR 5.22, 95% CI 1.08–25.3, *p* = 0.040).

Table 3 shows the multivariate analysis results. Lower baseline serum albumin levels were significantly associated with death or OLT in model 1 (intermediate vs. high albumin group: HR 3.51, 95% CI 1.85–6.63, *p* < 0.001; low vs. high albumin group: HR 23.3, 95% CI 8.42–64.6, *p* < 0.001), model 2 (intermediate vs. high albumin group: HR 3.26, 95% CI 1.72–6.20, *p* < 0.001; low vs. high albumin group: HR 23.6, 95% CI 8.51–65.6, *p <* 0.001), and model 3 (intermediate vs. high albumin group: HR 3.33, 95% CI 1.74–6.38, *p <* 0.001; low vs. high albumin group: HR 23.0, 95% CI 8.21–64.3, *p <* 0.001) (Table 3). Lower baseline serum albumin levels were also significantly associated with liver-related death in model 1 (intermediate vs. high albumin group: HR 11.4, 95% CI 1.85–6.63, *p <* 0.001; low vs. high albumin group: HR 99.8, 95% CI 24.1–412, *p <* 0.001). Older age (≥65 y) and male sex were significantly associated with death or OLT in all models (Appendix A). BMI.

## 4. Discussion

Our retrospective cohort study yielded at least two major findings. First, clinical outcomes of NAFLD could be stratified by the serum albumin level at the time of liver biopsy. Second, the prognostic utility of baseline serum albumin level for predicting the outcome of death or OLT remained significant after adjusting for age, sex, presence of histologic NASH, DM, and fibrosis stage, thus highlighting the importance of aggressive efforts to provide early nutritional intervention.

Hypoalbuminemia is a well-known risk factor for mortality and other clinically important adverse outcomes in a variety of patient populations [6,9]. Hypoalbuminemia has also been associated with poorer prognosis for individuals with liver disease, including NAFLD. For example, Kawanaka et al. followed 489 patients with biopsy-proven NAFLD for 1–22.2 years [21] and found that patients with an albumin level < 3.5 g/dL, platelet counts < 150 × 10^9^ /L, and type IV collagen 7S levels ≥ 5 ng/mL indicate a poor prognosis. In particular, the 10-year survival rate was only 43% in patients presenting with all three factors. They concluded that albumin is reported to be one of the most important prognostic factors [21]. Vilar-Gomez et al. followed 458 patients with biopsy-proven NAFLD and bridging fibrosis (F3, *n* = 159) or compensated cirrhosis (CTP score of A5, *n* = 222; CTP score of A6, *n* = 77) for 5.5 years (range, 2.7–8.2 years) [22]. The transplantation-free survival rate at 10 years was higher in patients with F3 fibrosis (94%; 95% CI 86%–99%) than in those with cirrhosis and a CTP score of A5 (74%; 95% CI 61%–89%) or A6 (17%; 95% CI 6%–29%). Importantly, lower albumin levels (3.0–3.5 g/dL) would explain the main differences observed between patients with a CTP score of A5 and those with a CTP score of A6 [22]. In our analysis, the hazard risk of all clinical events except HCC was 20–80 times higher in the low albumin group than in the high albumin group by Cox proportional hazard regression analysis (Figure 2); these results are consistent with those of previous reports. As mentioned above, NFS, which includes albumin as a variable, is useful for predicting the prognosis of NAFLD [11]. Furthermore, the Model for End-Stage Liver Disease (MELD), which is a known reliable predictor of short-term survival in patients with end-stage liver disease [23], was recently updated to MELD 3.0 to improve the accuracy of mortality prediction. This optimized version takes into account new variables, including serum albumin [23].

Branched-chain amino acids (BCAAs) are a group of essential amino acids consisting of valine, leucine, and isoleucine [8]. BCAA supplementation was originally proposed as a strategy to normalize amino acid profiles and nutritional status. However, large-scale, multicenter, randomized, double-blinded, controlled trials performed in Italy and Japan demonstrated that BCAA supplementation improves not only nutritional status but also prognosis and quality of life in patients with liver cirrhosis [8,24]. Specifically, a study from Japan revealed that serum albumin levels increased significantly after 2 years of oral BCAA administration [24]. The 2019 European Society for Clinical Nutrition and Metabolism guideline on clinical nutrition in liver disease recommends long-term oral BCAA supplements (0.25 g·kg^−1^·d^−1^) for patients with advanced cirrhosis to improve event-free survival or quality of life [25]. In our cohort, the percentage of patients with advanced fibrosis in the intermediate albumin group (albumin 3.5–4.0 g/dL) was only 25% (74/295) (Table 1). Moreover, the onset of clinically important events in this group often occurred approximately 5 years after the start of observation, which tended to be earlier than the time of onset in the high albumin group (Figure 1). In consideration of previous reports that patients with greater declines in serum albumin during follow-up have a poorer prognosis [26], BCAA administration to the intermediate albumin group may be beneficial to improve the prognosis of patients with NAFLD. However, the following points should be noted: (1) the inclusion criteria for BCAA use in patients with cirrhosis in the aforementioned Japanese study [24] defined a low albumin concentration as ≤3.5 g/dL, so its usefulness in patients with NAFLD, including those without cirrhosis, remains unclear, and (2) in most countries, oral BCAA supplements are not reimbursed, and the combination of cost and poor palatability may affect compliance [27].

In many aspects, BCAA supplementation was often thought to be beneficial to energy expenditure. However, increased circulating levels of BCAA are linked to obesity and insulin resistance/diabetes, and long time feeding of high BCAA diets shortens life span via obesity and metabolic abnormalities [28].

De Bandt JP et al. described an excellent review on BCAAs and insulin resistance [29]. Chronically elevated plasma levels of BCAAs may be a predictive marker for the risk of developing insulin resistance and ultimately type 2 DM. They concluded that under conditions of amino acid abundance, competition between amino acids and glucose occurs, adversely affecting glucose. It may indicate that the organism’s ability to store amino acids is limited. The limit is reached as soon as the protein pool is restored. Therefore, the priority is the removal of excess amino acids and conservation of glucose [29].

Solon-Biet SM et al. conducted a detailed study using a mouse model of various BCAA concentrations with key nutrients fixed at isocaloric. They reported that long-term exposure to high BCAA diets leads to hyperphagia, obesity and reduced lifespan [30]. Additionally, they found that elevated relative amounts of BCAAs and other amino acids (especially tryptophan and threonine) in the diet caused increased appetite, which was associated with central serotonin depletion.

Recent studies clarified the association between BCAAs and brown adipose tissue (BAT). BAT is a thermogenic organ that is effectively recruited on acute and chronic cold exposure. BAT primary source of energy for thermogenesis is its own triglyceride (TG) content, with glucose and amino acids contributing to rapid intracellular TG repletion. Glucose, BCAA, glutamate, and other sources of energy contribute mainly to drive de novo lipogenesis and glycerol synthesis that are essential to replete intracellular triglycerides and to sustain the very high rate of TG/nonesterified fatty acid cycling necessary for brown adipose thermogenesis [31]. Recently, Yoneshiro et al. reported that BAT in mice and humans actively utilizes BCAAs in mitochondria for thermogenesis and promotes systemic BCAA clearance during cold exposure. In contrast, BAT-specific defects in BCAA catabolism attenuate systemic BCAA clearance, BAT fuel oxidation, and thermogenesis, leading to diet-induced obesity and glucose intolerance. BCAA catabolism in BAT is mediated by SLC25A44 which transports BCAAs to the mitochondria [28]. These results suggest that BAT functions as an important filter to control metabolism; BCAA clearance via SLC25A44 contributes to improved metabolic health.

Taken together, we do not know whether administration of BCAAs to NAFLD patients with hypoalbuminemia increases albumin levels or prolongs prognosis. To begin with, the relationship between BCAA imbalance, skeletal muscle mass, and sarcopenia in patients with NAFLD with advanced liver fibrosis is still unknown, and further studies are needed to clarify these problems. For example, we believe that retrospective observational studies should examine whether BCAA imbalance is corrected and changes in body composition and prognostic factors in NAFLD patients treated with BCAAs.

Long-term administration of albumin can modify the course of decompensated cirrhosis by reducing the onset of new complications, improving the quality of life, and probably providing survival benefits [4,5]. There is, however, a need to rationalize the use of albumin therapy in different types of liver disease and stages of cirrhosis and to determine the optimum dose, duration, and frequency of albumin in each situation. Hypoalbuminemia can develop during acute stress, major operations, trauma, or infection and does not always require albumin replacement [4]. Indeed, it was estimated that 40%–90% of albumin prescriptions are unjustifiably given for correcting hypoalbuminemia per se, without considering the underlying disease process [4].

The main strengths of our study were the large number of included patients; confirmation of all cases of NAFLD by liver biopsy; grading and staging of liver biopsies by a single, experienced liver pathologist, thereby avoiding inter-observer variability; and the use of widely accepted scoring systems, including the FLIP algorithm, to grade and stage liver biopsy features [20].

This study also had some limitations, most of which are inherent to retrospective studies, including the absence of a specific treatment protocol, lack of follow-up endoscopic evaluations, and lack of imaging results in patients without cirrhosis. Thus, the number of liver-related events was possibly underestimated. In addition, the small number of the patients in the albumin lower group (*n* = 28) makes the data unstable, with a wide 95% CI for hazard ratios. Additionally, the follow-up period was shorter than in some other studies [32,33,34,35]. Nevertheless, this study was not meant to be a clinical trial; the goal was to analyze real-world data from patients who underwent liver biopsy.

## 5. Conclusions

In conclusion, baseline albumin levels allow appropriate prediction of patients with NAFLD at higher risk of developing death or OLT, liver-related death, HCC, decompensated cirrhosis, and gastroesophageal varices. These results are relevant for patient counseling and early nutritional intervention (especially an albumin level < 4.0 g/dL) in Asian patients with NAFLD.

## Figures and Tables

**Figure 1 nutrients-15-02014-f001:**
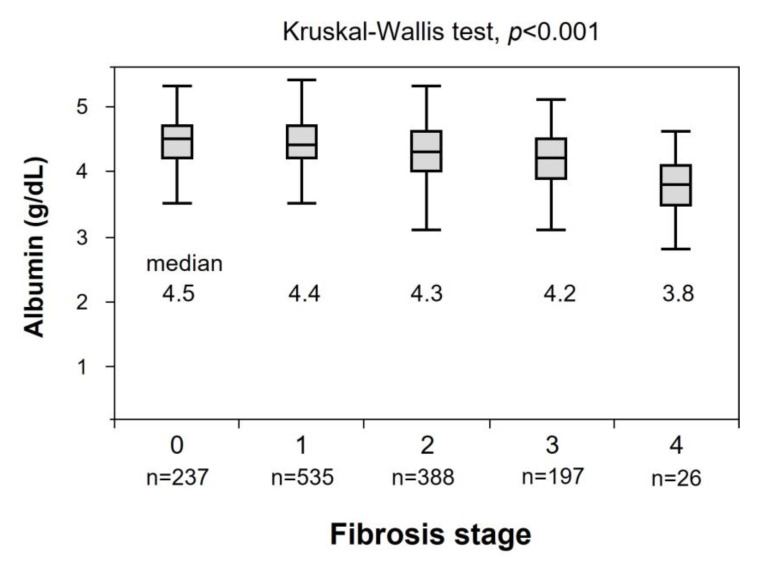
The relationship between fibrosis stage and albumin levels.

**Figure 2 nutrients-15-02014-f002:**
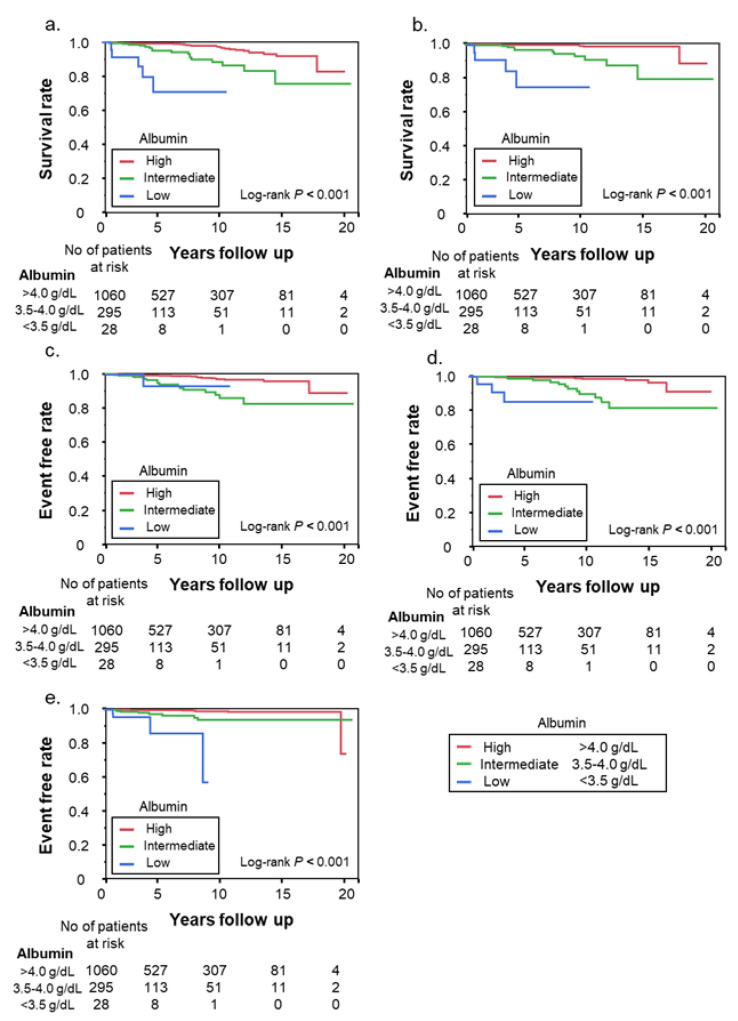
The relationship between fibrosis stage and albumin levels. (**a**) Death or OLT, (**b**) liver-related death, (**c**) hepatocellular carcinoma, (**d**) decompensated cirrhosis, (**e**) gastroesophageal varices.

**Figure 3 nutrients-15-02014-f003:**
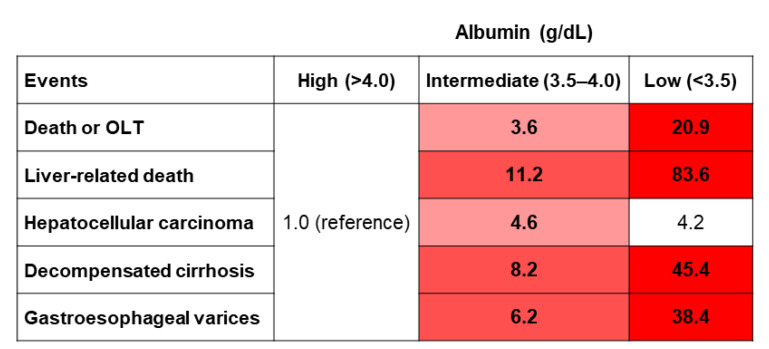
Heatmap of univariate unadjusted hazard risks of clinically important outcomes.

**Table 1 nutrients-15-02014-t001:** Clinical and demographic patient characteristics.

			Albumin (g/dL)	
Variables	Total N = 1383	High (>4.0) *n* = 1060	Intermediate (3.5–4.0) *n* = 295	Low (<3.5) *n* = 28
**Sex, No. (%)**				
Female	790 (57.1)	566 (53.4)	203 (68.8) ***	21 (75.0) *
Male	593 (42.9)	494 (46.6)	92 (31.2)	7 (25.0)
Age, mean (SD), y	54.6 (14.3)	53.0 (14.4)	59.8 (12.4) ***	60.5 (12.4) **
Age ≥ 65 y, No. (%)	388 (28.1)	257 (24.2)	120 (40.7) ***	11 (39.3)
BMI, mean (SD), kg/m^2^	27.9 (4.7)	27.8 (4.5)	28.2 (5.2)	29.8 (4.8) *
**Medical history, No. (%)**				
Hypertension	581 (42.0)	427 (40.3)	140 (47.5) *	14 (50.0)
DM	498 (36.0)	339 (32.0)	145 (49.2) ***	14 (50.0)
Dyslipidemia	793 (57.3)	617 (58.2)	163 (55.3)	13 (46.4)
**Laboratory data, mean (SD)**				
Albumin, g/dL	4.3 (0.4)	4.5 (0.3)	3.8 (0.2) ***	3.1 (0.6) ***
AST, U/L	61.2 (39.7)	58.9 (36.9)	67.6 (43.7) **	81.7 (73.6) **
ALT, U/L	88.3 (61.1)	90.5 (61.5)	81.8 (59.5) *	71.2 (56.2)
GGT, U/L	88.8 (96.3)	87.1 (88.2)	91.6 (114)	127 (164) *
Platelet count,×109/L	220 (70.3)	226 (67.3)	202 (73.2) **	156 (86.7) **
FIB-4 index	2.01 (1.61)	1.74 (1.27)	2.71 (1.94) ***	4.97 (3.30) ***
NFS	–1.57 (1.72)	–1.93 (1.56)	–0.51 (1.59) ***	1.14 (2.00) ***
**Liver histology, No. (%)**				
NASH (yes)	925 (66.9)	693 (65.4)	216 (73.2) *	18 (64.3)
Fibrosis stage				
1/2	1160 (83.9)	926 (87.4)	221 (74.9) ***	13 (46.4) ***
3/4	223 (16.1)	134 (12.6)	74 (25.1)	15 (53.6)

ALT, alanine aminotransferase; AST, aspartate aminotransferase; BMI, body mass index; DM, diabetes mellitus; FIB-4, fibrosis-4; GGT, gamma-glutamyltransferase; NASH, non-alcoholic steatohepatitis; NFS, NAFLD fibrosis score; SD, standard deviation. * *p* < 0.05, ** *p* < 0.01, *** *p* < 0.001, compared with the high albumin group.

**Table 2 nutrients-15-02014-t002:** Clinically important patient outcomes.

	Albumin (g/dL)			
	Total N = 1383	High (>4.0) N = 1060	Intermediate (3.5–4.0) n = 295	Low (<3.5) N = 28
**Death or OLT**	46	24	17	5
**Liver-related death**	20	2	11	4
**Liver-related events**				
Hepatocellular carcinoma	36	18	17	1
Decompensated cirrhosis	23	8	12	3
Gastroesophageal varices	17	6	8	3

OLT: orthotopic liver transplantation.

**Table 3 nutrients-15-02014-t003:** Multivariate adjusted hazard ratios for death or OLT and liver-related death.

		Death or OLT		Liver-Related Death	
	Albumin	HR (95% CI)	*p* Value	HR (95% CI)	*p* Value
**Model 1**	High	1 (reference)		1 (reference)	
	Intermediate	3.51 (1.85–6.63)	**<0.001**	11.4 (1.85–6.63)	**<0.001**
	Low	23.3 (8.42–64.6)	**<0.001**	99.8 (24.1–412)	**<0.001**
**Model 2**	High	1 (reference)			
	Intermediate	3.26 (1.72–6.20)	**<0.001**		
	Low	23.6 (8.51–65.6)	**<0.001**		
**Model 3**	High	1 (reference)			
	Intermediate	3.33 (1.74–6.38)	**<0.001**		
	Low	23.0 (8.21–64.3)	**<0.001**		

Significant results are highlighted in bold. CI, confidence interval; OLT, orthotopic liver transplantation. Model 1 includes age and sex. Model 2 includes variables from model 1 plus non-alcoholic steatohepatitis (based on fatty liver inhibition of progression algorithm) and fibrosis stage. Model 3 includes variables from model 1 plus diabetes mellitus and BMI. High, >4.0 g/dL; Intermediate, 3.5–4.0 g/dL; Low, <3.5 g/dL.

## Data Availability

The data presented in this study are available on request from the corresponding author. The data are not publicly available due to ethical aspects.

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
