# Peer review of "Association of Serum Albumin Levels and Long-Term Prognosis in Patients with Biopsy-Confirmed Nonalcoholic Fatty Liver Disease"

_nutrients, 2023, doi:10.3390/nu15092014_

Round 1

Reviewer 1 Report

Hirokazu Takahashi et al. are investigating the relationship between baseline serum albumin levels and long-term prognosis of NAFLD. Please find below major and minor comments which the reviewer think needs to be addressed to strengthen the manuscript and the study.

The study is a sub-analysis of the CLIONE study already published (https://doi.org/10.1016/j.cgh.2022.01.002). Therefore, the authors need to be careful to avoid copying whole text passages from their previous manuscript, without reference. Please rewrite the respective sections. (e.g. line 73-77, line 112-116, line 123-125, …).

Moreover, the authors mentioning that methodology for establishing the cohort are already published in the previous publication. However, there are still some methodological information missing, which should be included:

-Patients with biopsy proven NAFLD were extracted from the database of Japan Study Group of NAFLD, please provide inclusion and exclusion criteria, age? sex?, …?

-line 123: how many patients were excluded from the analysis? How many patients with excessive alcohol intake and all the other criteria? In line 103 1383 patients were enrolled and in table 1 all of them were analysed?

Number of patients with low albumin concentration is rather low and might influence the outcome, please comment on that.

Patients with low albumin were older and had a significantly higher BMI. As BMI is a risk factor for liver-related death and events this reviewer suggests to also include this within the analyis?

Figure 2: If possible, please include the headings of each single figure above the figures instead of using the figure legends.

Author Response

Manuscript ID: nutrients-2309074

Prof. Dr. Maria Luz Fernandez and Prof. Dr. Lluis Serra-Majem

Editor-in-Chief

Nutrients

Association of Serum Albumin Levels and Long-term Prognosis in Patients with Biopsy-Confirmed Nonalcoholic Fatty Liver Disease

Response to the comments of reviewers

We thank the editor and reviewers for the positive assessment of our manuscript and for identifying areas that required correction and /or modification. The corrected/modified text in the revised manuscript is indicated in red. All line numbers mentioned in the response to each comment refer to the numbers that appear at the left margin of the text in the revised manuscript. We really appreciate the reviewers for the comments to improve our manuscript.

Thank you for your kind consideration.

Yours sincerely,

Hideki Fujii

Department of Hepatology, Osaka Metropolitan University Graduate School of Medicine, 1-4-3 Asahimachi, Abeno-ku, Osaka 545-8585, Japan

Tel: +81-6-6645-3905

E-mail: rolahideki@omu.ac.jp.

Reviewer: 1

Comments to the Author

The study is a sub-analysis of the CLIONE study already published (https://doi.org/10.1016/j.cgh.2022.01.002). Therefore, the authors need to be careful to avoid copying whole text passages from their previous manuscript, without reference. Please rewrite the respective sections. (e.g. line 73-77, line 112-116, line 123-125, …).

              Thank you for the valuable comments. We rewrote the respective sections.

Revised Manuscript page 2, line 73- line 77:

NAFLD is a complication of obesity, and some patients are thought to progress to nonalcoholic steatohepatitis (NASH), which is characterized histologically by the presence of steatosis, inflammation, and hepatocellular ballooning with or without fi-brosis, ultimately leading to cirrhosis, hepatocellular carcinoma (HCC), and death.

Revised Manuscript page 2, line 102- page 3, line 107:

The method used to establish the cohort and the primary findings of the CLIONE study have been described elsewhere [13]. This investigation was conducted in accordance with the 1964 Declaration of Helsinki and was authorized by the institutional review board of Saga University Hospital (approval no. 2020-04-R-02, June 30, 2020). Informed consent was not deemed necessary since the study employed pre-existing data.

Revised Manuscript page 3, line 112- line 115:

In this study, we used a database from the Japan Study Group of NAFLD (JSG-NAFLD) to obtain information regarding patients with biopsy-proven NAFLD [13]. All data were managed using REDCap electronic data capture tools, hosted at the Osaka Metropolitan University.

Revised Manuscript page 3, line 140- line 144:

Ultrasound-assisted percutaneous liver biopsies were performed. Liver sections were embedded in paraffin after being fixed in formalin and stained with either he-matoxylin and eosin or azan. These sections were then sent to an experienced pathologist (S.A.) at Saga University for central evaluation, who was unaware of the patients' clinical and laboratory records.

Moreover, the authors mentioning that methodology for establishing the cohort are already published in the previous publication. However, there are still some methodological information missing, which should be included:

              Thank you for the valuable comments. We added the detail of the methodological information below.

Revised Manuscript page 3, line 118- line 125

We identified 1,760 patients who underwent liver biopsy for suspected fatty liver between December 1, 1994 and December 31, 2020 [13]. The exclusion criteria were a history of other hepatic diseases, such as viral hepatitis and a history of alcohol abuse (defined as > 30 g/day in men; >20 g/day in women). The inclusion criteria were 1) positive serology for HCV but negative for HCV-RNA and no history of treatment, and 2) burned-out NASH (no increased fat and fibrosis stage 4). We used a total of 1,398 patients in CLIONE study [13]. In this study, we also excluded 15 patients with missing serum

albumin data, and finally, enrolled 1,383 Japanese patients with biopsy-confirmed NAFLD.

Revised Manuscript page 3, line 118- line 125

Patient information was collected on an ongoing basis due to the fact that all events took place at the same facility where the patients were being treated. For hospitalizations, we documented the diagnosis given at the time of admission. The duration of follow-up was calculated as the period between the biopsy date and the most recent follow-up. In cases where the date or details of an event in the database were unclear, the lead investigator reviewed the facility records and made the necessary amendments.

-Patients with biopsy proven NAFLD were extracted from the database of Japan Study Group of NAFLD, please provide inclusion and exclusion criteria, age? sex?, …?

              Thank you for the valuable comments. We have listed the inclusion/exclusion criteria as follows.

Revised Manuscript page 3, line 121- line 125

The exclusion criteria were a history of other hepatic diseases, such as viral hepatitis and a history of alcohol abuse (defined as > 30 g/day in men; >20 g/day in women). The inclusion criteria were 1) positive serology for HCV but negative for HCV-RNA and no history of treatment, and 2) burned-out NASH (no increased fat and fibrosis stage 4). We used a total of 1,398 patients in CLIONE study [13]. In this study, we also excluded 15 patients with missing serum albumin data, and finally, enrolled 1,383 Japanese patients with biopsy-confirmed NAFLD.

Also, we described the characteristics of our cohort in Result section.

Revised Manuscript page 4, line 181- line 182

Baseline characteristics of the cohort (N=1,383) are presented in Table 1. Mean age was 54.6 years, mean BMI was 27.9 kg/m2, and females accounted for 57.1% (n=790) of the cohort.

-line 123: how many patients were excluded from the analysis? How many patients with excessive alcohol intake and all the other criteria? In line 103 1383 patients were enrolled and in table 1 all of them were analysed?

              As described in the revised manuscript, the CLIONE study analyzed 1,398 patients, excluding heavy drinkers and others, from a total of 1,760 patients with suspected fatty liver. Of these, 15 patients with no serum albumin data were excluded, making a total of 1,383 subjects for this subanalysis.

Revised Manuscript page 3, line 118- line 125

This sentence has been quoted before and will be omitted.

Number of patients with low albumin concentration is rather low and might influence the outcome, please comment on that.

              Thank you for the valuable comments. We agree with the reviewers. According to the reviewer’s suggestion, we have added the comment as limitation.

Revised Manuscript page 9, line 338- line 340

In addition, the small number of the patients in the albumin lower group (n=28) makes the data unstable, with a wide 95% CI for hazard ratios.

Patients with low albumin were older and had a significantly higher BMI. As BMI is a risk factor for liver-related death and events this reviewer suggests to also include this within the analyis?

              This is an important point. We also are interested in lean NAFLD. First, we created a Kaplan-Meier curve with a BMI cutoff of 23 and 25 (Appendix Figure 1). In
both cases, there was no significant difference in survival.

Overall mortality stratified according to BMI. a. cutoff BMI 25 kg/m2, b. cutoff BMI 23 kg/m2.

We also estimated univariate HR using Cox proportional hazard regression analysis. The univariate hazard ratio of BMI (≥25 kg/m2) for all-cause mortality was 0.98 (0.52-1.87), P=0.96. The hazard ratio for liver-related death was 2.16 (0.63-7.40), P=0.22 (Revised Supplementary Table 1). Furthermore, BMI was not extracted as a significant factor in the multivariate analysis (Revised Supplementary Table 2). These results indicate that, at least in our cohort, BMI is not a variable involved in all-cause or liver-related mortality.

Figure 2: If possible, please include the headings of each single figure above the figures instead of using the figure legends.

              Thank you for the valuable comments. We added the heading.

Reviewer 2 Report

Takahashi and colleagues conducted a compelling retrospective study examining the relationship between albumin levels and liver-associated outcomes. The study findings revealed that lower albumin levels were linked to a higher risk of liver-related outcomes, regardless of age or sex. However, the median follow-up duration of 4.5 years may not have been sufficient to evaluate liver-related outcomes, especially for those with F0-F2, which constituted the majority of the study population. Despite this limitation, the study was well-conducted, with a sufficient sample size and reliable histologic assessment using liver biopsy. The study results were also concisely described. I have only a few comments to address, which are listed below.

Comments

1.     The selection of variables for the multivariate analysis is quite understandable. However, it may be worthwhile to consider including BMI as a covariate for the analysis, as obesity is known to have an impact on liver-related outcomes.

2.     There have been multiple reports indicating that lean-NAFLD may lead to worse outcomes than non-lean NAFLD. Therefore, it would be more engaging for readers if the authors compared the outcomes of both lean and non-lean NAFLD groups.

3.     Regarding Table 3, where you mentioned that 'stage' was included in model 2, I would like to clarify if you are referring to fibrosis stage?

Author Response

(The authors gave the same response as above.)

Round 2

Reviewer 1 Report

no further comments